# The Prevention of Maternal Phenylketonuria (PKU) Syndrome: The Development and Evaluation of a Specific Training Program

**DOI:** 10.3390/nu16234111

**Published:** 2024-11-28

**Authors:** Carmen Rohde, Alena Gerlinde Thiele, Anne Tomm, Dinah Lier, Kathrin Eschrich, Christoph Baerwald, Skadi Beblo

**Affiliations:** 1Department of Women and Child Health, Hospital for Children and Adolescents, University Hospitals, University of Leipzig, Liebigstraße 20 a, 04103 Leipzig, Germany; alena.thiele@medizin.uni-leipzig.de (A.G.T.); anne.tomm@medizin.uni-leipzig.de (A.T.); skadi.beblo@medizin.uni-leipzig.de (S.B.); 2Department of Pediatrics, Klinikum am Steinenberg, Steinenbergstr. 31, 72764 Reutlingen, Germany; lier_d@klin-rt.de; 3Outpatient Clinic for Obstetrics and Gynecology, Eilenburger Straße 59, 04103 Leipzig, Germany; kathrineschrich@gmx.li; 4Faculty of Medicine, University Hospital for Internal Medicine, University Hospital, University of Leipzig, Liebigstraße 20, 04103 Leipzig, Germany; christoph.baerwald@medizin.uni-leipzig.de; 5Center for Rare Diseases, University Hospital, University of Leipzig, 04103 Leipzig, Germany

**Keywords:** maternal PKU syndrome, prevention, training program, pregnancy in phenylketonuria (PKU), preconceptional training

## Abstract

Background: Maternal phenylketonuria (PKU) syndrome, leading to severe psychomotor retardation, microcephaly, cardiac defects and undergrowth, affects the unborn children of mothers with PKU with insufficient metabolic control during pregnancy. To improve long-term outcomes, a specific prevention program was developed. Methods: We designed a group training program for young women with PKU (>14 years) and their partners. Knowledge regarding PKU therapy and pregnancy was evaluated by a specifically developed multiple-choice questionnaire. In addition, scores of anxiety and depression were evaluated. Results: Patients (*n* = 20) and their partners (*n* = 13) significantly improved their knowledge after participation (correct answers: patients—86% vs. 90%, *p* = 0.003; partners—78% vs. 89%, *p* = 0.012). Females significantly improved their knowledge about diet (90% vs. 100%, *p* = 0.035) and metabolism (91% vs. 100%, *p* = 0.016), but not concerning gynecological topics. Patients’ median depression scores were within the normative range, with a slight decrease over time (6 points vs. 4 points, *p* = 0.836). Patients’ anxiety score remained stable over time (5.5 vs. 5, *p* = 0.247). Of trained mothers with PKU, four pregnancies with ideal metabolic control and healthy offspring could be observed. Conclusion: We suggest the inclusion of specific training programs in the standard care of female young adults with PKU, including for their partners.

## 1. Introduction

Maternal phenylketonuria (PKU) syndrome is a severe embryo-fetopathy. It affects the unborn children of mothers with PKU with non-satisfactory metabolic control during pregnancy [1]. High phenylalanine (Phe) concentrations negatively influence organogenesis and growth, and the fetal brain appears especially sensitive. Typical signs of maternal PKU syndrome are growth retardation, microcephaly, congenital heart defects and severe intellectual disability [2]. Thus, the prevention of maternal PKU syndrome is one of the most important goals in the professional care of women with PKU of childbearing age. To achieve this, mothers need to maintain plasma/dried-blood Phe concentrations between 120 and 360 µmol/L at conception and during pregnancy [3,4,5,6]. This can be accomplished by a strict Phe-reduced diet and supplementation with Phe-free protein substitutes adapted to the special needs during pregnancy.

Adhering to the Phe-reduced diet is a major challenge at any age. During adolescence, patients may commonly relax the diet, since it is acceptable for Phe concentrations to rise from below 240–360 µmol/L (the recommended limit during childhood) to up to 600 µmol/L in adults [7]. However, many adolescents and adults relax the diet far beyond their respective recommendations [8]. The most commonly stated reason is the bad taste of the amino acid substitute, which needs to be taken several times daily to ensure an adequate supply of nutrients. Additionally, the high effort needed to integrate the diet into daily life is perceived as an obstacle [9]. Thus, shifting back to the strict childhood diet prior to and during pregnancy in order to reach a safe level of metabolic control requires unusually tight discipline and effort [10,11]. Even though education about maternal PKU syndrome is common practice at regular clinic visits, many patients do not put their knowledge into practice. In addition, anxiety and depression resulting from this chronic disease and the burden of treatment may distract patients with PKU from achieving good compliance, which in turn may again induce depression and anxiety [12]. A lack of education and uncertainty about maternal PKU may also drive anxiety and depression [13].

As a result, many female patients only consult metabolic centers after becoming pregnant, without prior planning. Consequently, the number of children showing signs of maternal PKU syndrome has increased in recent years [14].

To further improve long-term outcomes for children born by women with PKU, we developed a specific training program for potential future mothers. The program aims to empower women with PKU and their partners with respect to knowledge and understanding of PKU in general, PKU therapy, sexual education, and pregnancy with PKU. Based on the results of a questionnaire that we had specifically developed to study the experiences and needs of mothers with PKU, we were able to tailor this training program according to knowledge gaps and preferences identified by the respondents [15]. In the study presented here, we investigate whether the training program improves knowledge regarding PKU self-management, metabolism and pregnancy in general, in both women with PKU and their partners. In addition, we evaluate depression and anxiety scores in women with PKU before and after training.

## 2. Methods

### 2.1. Training Program

The metabolic center of Leipzig, Germany, designed a two-part (module A and module B) group training program for young women with PKU and their partners. Each module offered a one-day program, and the modules were taught at least 4 months apart. Participants could take part in one or both modules in sequential or reverse order. Each module was offered three times during the period from September 2019 to July 2022.

Module A covered the following topics: PKU therapy for girls and young women at a specialized pediatric metabolic center, sex education, fertility, contraception, background information about pregnancy in general, dietetics during pregnancy with PKU, group discussion with psychological supervision, and the personal reports of and an exchange of insights with experienced mothers with PKU.

The content of module B covered the following topics: PKU therapy for girls and young women at a specialized adult metabolic center, practical advice, recipes for the realization of the diet, diet calculation, group discussion with psychological supervision and, again, personal reports of pregnancies from experienced mothers with PKU.

All modules were taught in person. The program included lectures, workshops and group discussions.

The evaluation of the training program was approved as a multi-center study by the ethics committee of the University of Leipzig’s Medical Faculty (reference number 108/19-ek; dated 26 March 2019). It is registered in the DRKS (German clinical trial register) as well as on the WHO portal (DRKS00032704). Patients and their partners provided written informed consent to analyze their data in a pseudonymized manner.

### 2.2. Participants

All female patients with PKU of childbearing age (>14 years) cared for at our metabolic center in Leipzig, Germany, as well as from the metabolic centers in Dresden, Berlin, Halle and Jena, and their partners were invited to participate. Parents of girls with PKU older than 6 years served as a control group for the multiple-choice questionnaire. They did not participate in the training; however, all parents of girls with PKU are routinely taught concerning awareness of maternal PKU syndrome during regular clinical visits.

### 2.3. Data Collection

Before and 4–6 months after each module, the following data were collected.

Diet records over three days were evaluated with respect to protein and micronutrient supply.

Metabolic control was assessed based on the Phe concentrations in dried-blood samples obtained one to two years prior to and after the training, with up to 10 measurements, each.

The analysis of Phe in dried blood was performed by liquid chromatography/tandem mass spectrometry (LC-MS/MS) as previously described [16].

Knowledge regarding PKU therapy and pregnancy was evaluated by a specifically developed multiple-choice questionnaire with four response options, of which only one was correct. For module A, we presented 20 questions, and for module B, 12 questions were prepared, addressing three topics: diet; Phe/energy, amino acid, fat and carbohydrate metabolism; and pregnancy in general, as well as specific gynecological themes for women with PKU. Answers were evaluated for the total number of correct answers and according to the three topic areas separately. The translated questionnaire is provided in Appendix A.

Anxiety and depression were evaluated by the Beck Anxiety Inventory (BAI) [17] and the Beck Depression Inventory II (BDI-II) [18]. The BAI is a self-administered questionnaire of anxiety symptoms, retrospectively covering the previous week. It consists of 21 items, each describing a common symptom of anxiety (for example “numbness, unsteady nerves”) on a 4-point Likert scale from 0 to 3. The sum of all points represents the total score, and anxiety level can be interpreted as follows: 0–7 = low anxiety; 8–15 = mild anxiety; 16–25 = moderate anxiety; and 26 and above = clinically relevant anxiety [17]. The BDI-II is a 21-item self-report inventory measuring the severity of depressive symptoms in adolescents and adults, retrospectively covering the previous two weeks. Each of the 21 questions presents four different statements and asks respondents to select the option that best represents their current emotional state. Statements refer to depressive states in varying degrees of severity on a 4-point Likert scale from 0 to 3. The sum of all points, the total score, represents different levels of depressive symptoms: 0–8 = no depression; 9–13 = minimal depression; 14–19 = mild depression; 20–28 = moderate depression; 29–63 = severe depression [18].

COVID-19: To control for the possible impact of the SARSCoV2 pandemic [19] on questionnaire (BAI and BDI) answers, we asked all participating patients whether and to what degree they perceived the SARSCoV2 pandemic to have an influence on their daily life (response options: no; yes, a positive influence; or yes, a negative influence).

### 2.4. Statistics

All statistical procedures were performed using SPSS 29 [20].

Diet: Three-day diet records were evaluated using the food composition database DIÄT-2000 Professional [21]. Nutrient intake was calculated as the mean of three days.

Metabolic control: The median of mean Phe obtained from dried-blood concentrations one to two years prior to enrollment was compared to the median of mean Phe obtained one to two years after training using the Wilcoxon test. In addition, a comparison of the metabolic control, expressed by dried-blood Phe concentration, of participating patients with that of those who were invited but did not participate was performed using the Kruskal–Wallis test.

Knowledge: Since patients could participate in modules A, B or both, we calculated the percentage of correct answers rather than the sum score. The Wilcoxon test was used to compare results before and after the training modules, for patients and partners separately. The Kruskal–Wallis test was used for comparison with the control group.

Anxiety and depression: Scores on the questionnaires at baseline and after participation in the training program were compared (Wilcoxon test), independent of if one or both modules were attended. Correlations of dried-blood Phe concentrations prior to and after the program were evaluated with Spearman’s rank correlation coefficient.

## 3. Results

A total of 20 patients and 13 partners accepted the invitation for the training program (Table 1). Four women did not have a partner at the time of the training, one woman had already experienced a pregnancy and her partner felt well informed, and two partners could not participate due to professional obligations.

### 3.1. Diet

Diet records before and after the training program were available from ten patients. All but three patients consumed a Phe-free protein substitute according to current recommendations [22]. One of these patients restarted the supplementation after participation in the first module of the training, one was using BH4 and did not need supplementation at the time of study participation and one patient continuously refused intake. All available diet records indicating the use of a protein substitute confirmed the sufficient protein intake for PKU of 140% of the recommendation for a healthy population (mean of 130%, median of 124%, and range of 96–164%) [22,23]. They also met the recommended intake [23] of folic acid, vitamin B12, vitamin A, vitamin D, calcium, iodine, iron and zinc. The three patients who were not consuming a protein substitute could not meet the micronutrient recommendations; their protein intake met the recommendation for healthy women (mean of 106%, median of 115%, and range of 60–146).

### 3.2. Metabolic Control

Metabolic control throughout the study period is shown in Table 1. The median of mean dried-blood Phe concentrations was within the recommended range for adult PKU patients (120–600 µmol/L) [22] and did not change over time (before vs. after training, *p* = 0.502). However, the variability in metabolic control was pronounced: one third of the patients only showed values above the recommended range, whereas another third strictly followed current recommendations. No differences in metabolic control between participants and invited but not participating patients could be revealed. This was true for median Phe concentrations (prior to training, *p* = 0.249; after training, *p* = 0.279) as well as for the proportion of Phe concentrations above the recommended range (prior to training, *p* = 0.227; after training, *p* = 0.114).

### 3.3. Knowledge

Before the training, patients with PKU and their partners showed good basic knowledge about the diagnosis of PKU in general as well as dietary treatment strategies and the achievement of good metabolic control. In addition, females’ basic knowledge about gynecological topics was better in comparison to their partners’.

Both females with PKU and their partners significantly improved their knowledge after participation in the training program.

Females significantly improved their knowledge about diet and metabolism, but not concerning specific gynecological topics. In contrast, the partners achieved a significant improvement in their total score, which could not be attributed to the single topics (knowledge about diet, metabolism and pregnancy). Controls (parents of school-age girls with PKU) showed slightly less knowledge in all areas before the training program compared to participating females, but better knowledge than partners. The differences did not reach significance.

After the training, partners could make up their knowledge gap. In addition, they performed better compared to the control group regarding questions about diet and metabolism. Probably due to the small number of participants, the differences did not reach significance. The exact data are given in Table 2.

### 3.4. Depression and Anxiety

As the training program was partly performed during the SARSCoV2 pandemic, a possible influence on depression and anxiety scores could be assumed [24]. This applied to 11 patients; the other 9 had already finished the program before the pandemic. Four (36%) of them indicated a negative influence, two (18%) of them indicated no influence and five (45%) even indicated a positive influence. Overall, we could not detect a specific influence on the questionnaire results.

At baseline and after the training, patients’ median BDI scores were within the normative range compared to a healthy population. A total of 4 (20%) out of the 20 patients were classified as “mildly” or “moderately” depressed before the training; 3 of them showed lower scores after participation. No systematic influence of the COVID-19 pandemic on BDI scores could be revealed. Of the three patients with increasing BDI scores over time, two completed their training during the pandemic restrictions. However, the patient with the most severe BDI score completed the training long before any sign of or restrictions surrounding the SARSCoV2 pandemic.

On average, patients showed low anxiety, as assessed by the BAI score before training; the score remained unchanged over the training period. At baseline, 4 (20%) out of the 20 patients presented a moderate/clinically relevant score of anxiety, but all of them showed better results afterwards (Table 3). Compared to the general population, the anxiety score of the patients is elevated.

A correlation between metabolic control and depression or anxiety scores could not be detected in our patient group (Table 4).

## 4. Discussion

The prevention of maternal PKU syndrome is one of the most important goals in the care of adult female patients with PKU. Many young women with PKU are not entirely aware of the risk of an unplanned pregnancy [25]. This seems particularly problematic in connection with a relaxed diet, poor compliance [26] and increasing tolerance to elevated Phe levels from adolescence onwards [22].

With the development of a specific training program for women with PKU of childbearing age, we pursued the goal of first determining general knowledge in this field and secondly preventing or diminishing the occurrence of maternal PKU syndrome in the future. To evaluate its effectiveness, the first 20 participating PKU patients, as well as their partners, completed a multiple-choice questionnaire covering essential topics within PKU therapy throughout pregnancy as well as general knowledge about sex education, pregnancy and contraception before and 4 months after the completion of training. All topics are of significant importance, and especially when taught within the training program. The training program was evaluated and rated as very good by all participants (3.7 points out of a total of 4).

Fortunately, as a group, the patients showed normative values regarding depression or anxiety. The data are comparable to the standard sample of metabolically healthy people [17,18]. However, the variability in the results is high, although a connection to metabolic control could not be established. Anxiety or depression data in PKU are scarce. Available data show diverse results [27,28]. 

We previously reported a very good average quality-of-life in children and adolescents with PKU, also varying widely, but with a direct correlation with the need of dietary treatment [29]. The data from Klimek et al. confirm the good social integration in the living situations of adult patients with PKU who received early treatment [30]. However, in another study at our center, using self-report and parent-report questionnaires, we showed that female adolescents are especially at risk of suffering from emotional problems [31]. In this respect, the BDI and BAI might not be specific enough to reveal important aspects of emotional wellbeing in this very special patient group.

General knowledge about PKU, sex education and contraception was good overall and comparable to the control group, even before the training program. This is all the more reassuring as three participants had already experienced a pregnancy (with healthy offspring). In this respect, they appeared to be quite well equipped. Nevertheless, they were able to significantly improve their knowledge, especially with regard to metabolic control and dietary management during pregnancy, as a result of the training.

Just like the patients, the participating partners benefited from the training program. The overall increase in knowledge was highest in this group, even if this was only significant for the comparison between the overall result before and after training, presumably due to the small number of participants. In an earlier study at our center, we were able to identify “support from the partner” as one of the most important supportive factors during pregnancy [15]. This support is only possible if the partner has a sound knowledge of the subject, as has been clearly demonstrated in studies of larger cohorts with other chronic diseases [32]. This is particularly important with regard to PKU, where a constant good metabolic control throughout pregnancy is extremely important. It has been shown that Phe fluctuations may have an especially negative impact on the fetus [6] and therefore should be avoided. While women with PKU themselves have to face and have been familiar with their disease and its management since early childhood, the partners usually have had no contact with a highly specialized dietary treatment before their relationship, including an exact calculation of daily protein intake, precise dosage and the intake of an amino acid mixture. The importance of daily management in PKU is well illustrated by the results from the analyzed diet records. The protein and micronutrient intake for all participants who provided us with diet records was only sufficient [23] if they took a protein substitute. The correlation between low micronutrient supply and low protein intake is known [33]. In preparation for a pregnancy, this fact must be discussed with the patient accordingly.

The metabolic control of the participating patients was good on average, but a large proportion of them showed dried-blood Phe values above the target range (120–600 µmol/L) [22]. This is due to the fact that most of the patients did not want to become pregnant during or right after the course. Over time, four of the trained women from different centers became pregnant. We were able to follow these pregnancies. In two of the women with PKU, all dried-blood Phe concentrations (n = 34 and n = 41) during pregnancy were within the target range (120–360 µmol/L); in the third and fourth pregnancy, only one measurement, out of more than 60, was slightly above the target range. All four women gave birth to healthy children. In comparison, only one invited but non-participating, non-trained patient across all participating centers became pregnant during the same time period. She delivered on term, but the baby presented low birth weight (2750 g), growth retardation and microcephaly. Metabolic control over time was acceptable; however, the frequency of Phe measurements and visits at the outpatient clinic was suboptimal, and supplementation with Phe-free amino acid mixture was insufficient. The experience of four consecutive metabolically ideal pregnancies after specific training is certainly exceptional, and we assume that our training has raised the patients’ awareness of the importance of good metabolic control by a stringent diet during pregnancy. This recent experience led us to implement the training program into regular care for PKU women at our center. In addition, we are currently offering to host the program at other centers within Germany.

## 5. Conclusions

The included women with PKU had a relatively good general knowledge about the implications of PKU for family planning. However, a specialized training program can improve knowledge and awareness in patients and their partners. This training could help reduce the prevalence of maternal PKU syndrome. We recommend including it in the standard of care for female adolescents and young adults with PKU and their partners.

## Figures and Tables

**Table 1 nutrients-16-04111-t001:** Patient characteristics and metabolic control over time.

Subjects	Median (Range)	*n*
Women with PKU (age in years) − Tetrahydrobiopterin therapy − Previous motherhood − Pregnancy during the course (planned before participation)	28 (16–35)	20
		2
		3
		2
School degree obtained by young women with PKU − Secondary school (10 years) − High school (12 years) − Unknown		
		8
		10
		2
Women with PKU participating in modules − Module A and B − Module A or B		
		11
		9
Diet records − Daily Phe intake before participation (mg) − Daily Phe intake after participation (mg)		10
	827 (450–3350)	
	991 (441–3150)	
Dried-blood Phe prior to * training − Phe in µmol/L − % values > recommended range (360 µmol/L) − No values > recommended range (360 µmol/L) − All values > recommended range (360 µmol/L)		14
	515 (250–845)	
	15 ** (0–100)	
		4
		5
Dried-blood Phe after * training − Phe in µmol/L − % values > recommended range (360 µmol/L) − No values > recommended range (360 µmol/L) − All values > recommended range (360 µmol/L)		15
	475 (109–953)	
	20 ** (0–100)	
		6
		4
Partners		13
Patients invited, but NOT participating		
Dried-blood Phe prior * to planned training − Phe in µmol/L − % values > recommended range (360 µmol/L) − No values > recommended range (360 µmol/L) − All values > recommended range (360 µmol/L)		22
	549 (219–1208)	
	50 ** (0–100)	
		3
		6

Patient characteristics regarding age, therapy, previous motherhood, participation in modules, diet records, metabolic control and participating partners. * Up to 10 measurements 1–2 years prior to/after training. ** Participants prior to training compared to non-participants, *p* = 0.249; participants after training compared to non-participants, *p* = 0.279.

**Table 2 nutrients-16-04111-t002:** Knowledge test.

		All Topics			Diet			Pregnancy			Metabolism	
	Before	After	*p*	Before	After	*p*	Before	After	*p*	Before	After	*p*
Correct answers (%)												
Patients	86	90	0.003	90	100	0.035	83	83	n.s.	91	100	0.016
Partners	78	89	0.012	80	92	n.s.	67	80	n.s.	86	95	n.s.
Controls	84	n.a.	n.a.	87	n.a.	n.a.	83	n.a.	n.a.	73	n.a.	n.a.
Comparison (*p*)												
Patients vs. partners	n.s.	n.s.		n.s.	n.s.		n.s.	n.s.		n.s.	n.s.	
Patients vs. controls	n.s.	0.005		n.s.	n.s.		n.s.	n.s.		n.s.	<0.01	
Partners vs. controls	n.s.	n.s.		n.s.	n.s.		n.s.	n.s.		n.s.	n.s.	

Comparison of multiple-choice questionnaire results before and after the training (Wilcoxon test). Comparison of knowledge tests of controls, patients and partners (Kruskal–Wallis). n.s. = not significant; n.a. = not applicable.

**Table 3 nutrients-16-04111-t003:** Depression and anxiety scores.

	Depression Score			Comparison		
	Before	After	Norm [18]	Before–Norm	After–Norm	Before–After
Median (range)	6 (0–27)	4 (0–29)	5 (not available)	0.532	0.429	0.856
0–8 = no depression % (*n*)	70 (14)	80 (16)	65			
9–13 = minimal depression % (*n*)	10 (2)	0 (0)	20			
14–19 = mild depression % (*n*)	15 (3)	10 (2)	5			
20–28 = moderate depression % (*n*)	5 (1)	5 (1)	5			
29–63 = severe depression % (*n*)	0 (0)	5 (1)	5			
	Anxiety Score			Comparison		
	Before	After	Norm [17]	Before–Norm	After–Norm	Before–After
Median (range)	5.5 (0–27)	5 (0–22)	1 (0–60)	0.421	0.557	0.247
0–7 = low anxiety % (*n*)	60 (12)	65 (13)	79			
8–15 = mild anxiety % (*n*)	15 (3)	30 (6)	16			
16–25 = moderate anxiety % (*n*)	15 (3)	5 (1)	4			
>25 = clinically relevant anxiety% (*n*)	10 (2)	0 (0)	2			

Comparison of the depression and anxiety scores of patients before and after the training (Wilcoxon test) and comparison to the normative sample (Mann–Whitney U).

**Table 4 nutrients-16-04111-t004:** Correlation between metabolic control and depression and anxiety score.

	Metabolic Control					
	Before Training		At Training		After Training	
	r	*p*	r	*p*	r	*p*
Depression score before training	0.056	0.862	0.200	0.532		
Depression score after training					0.508	0.920
Anxiety score before training	0.212	0.509	0.434	0.157		
Anxiety score after training					0.334	0.289

Depression and anxiety scores in correlation (Spearman) with metabolic control at the start, 1–2 years before and 1–2 years after the training of all patients with available laboratory data (*n* = 12).

## Data Availability

Availability of data and materials: the datasets generated and/or analyzed during the current study are not publicly available due to privacy protection of the patients, but are available from the corresponding author on reasonable request.

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
