# Peer review of "The Prevention of Maternal Phenylketonuria (PKU) Syndrome: The Development and Evaluation of a Specific Training Program"

_nutrients, 2024, doi:10.3390/nu16234111_

Round 1
Reviewer 1 Report
Comments and Suggestions for Authors
Thank you for asking me to review this paper on developing and evaluating a training programme for women with PKU preparing for pregnancy. It is an important piece of work as well managed phe levels during pregnancy is paramount for the safety of the baby. After doing a literature search, I cannot find another study evaluating mPKU training.
Here are some small comments for you to improve the paper.
Introduction
1) First line – Start the sentence with Maternal (no need for The The!)
2) Line 47 – for women with PKU of child bearing age – ‘of’ rather than ‘in’.
3) Line 50 – ‘phe-free protein’ – do you use GMP as well as amino acids?
4) Line 53 + 55 – ‘loosen the diet’ in English we would say ‘relax the diet’.
5) Line 62 – ‘praxis’ should be practice.
6) Line 63-65 – This needs to be written better – why would they have depression and anxiety to start with? Why does depression and anxiety ‘distract’ from achieving good compliance?
Materials and Methods
1) It would be good if you could explain more about the training here. For example was it in person or virtual? Were they all lectures or workshops?
2) Participants – can you make it clearer about the control group – did they do the training or not? Where had they been taught the awareness of mPKU syndrome?
3) Line 121 – when you say ‘metabolism in general’ do you mean phenylalanine metabolism? Metabolism is a big subject!
Results
1) Table 1 – Diet records – daily phe intake after participation – it says mg 4 – is that a mistake?
2) The little ‘2’ indicating participants characteristics – the little 2 made me think that it is the number squared! Could you use symbols instead of numbers to make it clearer?
3) 3.2 Diet – Why were 3 subjects not taking protein substitute? They were not doing the diet/they were on sapropterin/they were mild?? Can it be clearer?
4) 3.4 knowledge – line 205 – take out the word ‘could’.
5) 3.5 Depression and anxiety – line 230 figure 19 is in the middle of a word!
Discussion
1) Line 283 – sexuality – is this the right word – definition – ‘a person's identity in relation to the gender or genders to which they are typically attracted; sexual orientation’. Should it be sexual education?
2) Line 300 – confronted – means hostility – is that what you wanted to portray?
3) Will you continue to provide this training as part of care of women with PKU? Should it be provided by every service?
4) Line 334 – not very good English ‘such training could be a possible chance’. This training could help reduce the prevalence of maternal PKU syndrome.
Author Response
Dear reviewer,
Thank you very much for your very valuable comments! Below you find our corrections:
Introduction
1) First line – Start the sentence with Maternal (no need for The The!)
We erased the second The!
2) Line 47 – for women with PKU of child bearing age – ‘of’ rather than ‘in’.
We changed “of” in “in”.
3) Line 50 – ‘phe-free protein’ – do you use GMP as well as amino acids?
Yes!
4) Line 53 + 55 – ‘loosen the diet’ in English we would say ‘relax the diet’.
We changed “loosen” into "relax”
5) Line 62 – ‘praxis’ should be practice.
We changed “paxis” into “practice”
6) Line 63-65 – This needs to be written better – why would they have depression and anxiety to start with? Why does depression and anxiety ‘distract’ from achieving good compliance?
We changed the sentence “In addition, anxiety and depression may distract PKU patients from achieving good compliance, which in turn may lead to increased depression and anxiety.” to “. In addition, anxiety and depression resulting from this chronic disease and burden of treatment may distract PKU patients from achieving good compliance, which in turn again may induce depression and anxiety [12].”
Materials and Methods
1) It would be good if you could explain more about the training here. For example was it in person or virtual? Were they all lectures or workshops?
We added the sentence “All modules were taught in person. The program included lectures, workshops and group discussions”.
2) Participants – can you make it clearer about the control group – did they do the training or not? Where had they been taught the awareness of mPKU syndrome?
We changed the sentence “Parents of girls with PKU older than 6 years served as a control group for the multiple-choice questionnaire, as they had been taught for awareness of maternal PKU syndrome.” To “Parents of girls with PKU older than 6 years served as a control group for the multiple-choice questionnaire. They did not participate in the training, but had been taught for awareness of maternal PKU syndrome during regular clinical visits.”
3) Line 121 – when you say ‘metabolism in general’ do you mean phenylalanine metabolism? Metabolism is a big subject!
We changed “metabolism in general” into “, Phe-/energy- amino acid-, fat- and carbohydrate-metabolism”,
Results
1) Table 1 – Diet records – daily phe intake after participation – it says mg 4 – is that a mistake?
Yes, this is a mistake!
2) The little ‘2’ indicating participants characteristics – the little 2 made me think that it is the number squared! Could you use symbols instead of numbers to make it clearer?
We changed 1 and 2 into * and **.
3) 3.2 Diet – Why were 3 subjects not taking protein substitute? They were not doing the diet/they were on sapropterin/they were mild?? Can it be clearer?
We added to the sentence “All but three patients consumed a Phe-free protein substitute according current recommendations [21]. One patient restarted the supplementation after the training, two patients refused the intake continuously.” the following: “One of these restarted the supplementation after participation in the first module of the training, one was using BH4 and not needing supplementation at time of study participation and one patient refused the intake continuously.”
4) 3.4 knowledge – line 205 – take out the word ‘could’.
We changed the sentence to “Both, PKU females and partners significantly improved their knowledge after participation in the training program.”
5) 3.5 Depression and anxiety – line 230 figure 19 is in the middle of a word!
We erased the figure.
Discussion
1) Line 283 – sexuality – is this the right word – definition – ‘a person's identity in relation to the gender or genders to which they are typically attracted; sexual orientation’. Should it be sexual education?
Yes, we talk about sex education and changed the expression.
2) Line 300 – confronted – means hostility – is that what you wanted to portray?
We changed the sentence to “While women with PKU themselves have to face and are familiar with their disease and its management since early childhood….”
3) Will you continue to provide this training as part of care of women with PKU? Should it be provided by every service?
We added the sentence: “This recent experience led us to implement the training program into regular care for PKU women at our centre. In addition, we are currently offering to host the program at other centres within Germany.”
4) Line 334 – not very good English ‘such training could be a possible chance’. This training could help reduce the prevalence of maternal PKU syndrome.
We replaced our sentence by your suggested one.
Reviewer 2 Report
Comments and Suggestions for Authors
The study described in the manuscript is intended to increase awareness and knowledge regarding the importance of a well-controlled diet and micronutrient supplementation in patients with phenylketonuria to minimize the occurrence of maternal phenylketonuria syndrome. To accomplish this goal, the authors designed training for patients and their partners and utilized a multiple-choice questionnaire before and after the training to adequately measure the acquired knowledge. The study is well-designed, and the method is appropriate to measure the effectiveness of the training. The statistical method is suitable, and the results are clearly stated. The conclusions are supported by the results.
Minor critics:
1) Please delete one of the reference 14 labels in line 75 or change it to an additional reference.
2) Please delete reference 19 in line 230.
Author Response
Dear reviewer,
Thank you very much for your very valuable comments!
1) Please delete one of the reference 14 labels in line 75 or change it to an additional reference.
2) Please delete reference 19 in line 230.
We erased both reference numbers.